# Mechanical Incipient Fault Detection and Performance Analysis Using Adaptive Teager-VMD Method

Huipeng Li [1,2], Bo Xu [1,2,*], Fengxing Zhou [1,*] and Pu Huang [2]

1 School of Information Science and Engineering, Wuhan University of Science and Technology, Wuhan 430081, China
2 School of Physics and Electronic Information, Huanggang Normal University, Huanggang 438000, China
* Correspondence: xubo6496918@163.com (B.X.); zhoufengxing@wust.edu.cn (F.Z.)

**Abstract:** For large rotating machinery with low speed and heavy load, the incipient fault characteristics of rolling bearings are particularly weak, making it difficult to identify them effectively by direct signal processing methods. To resolve this issue, we propose a novel approach to detecting incipient fault features that combines signal energy enhancement and signal decomposition. First, the structure of a conventional Teager algorithm is modified to further increase the energy of the micro-impact component and hence the impact amplitude. Then, a kind of composite chaotic mapping is constructed to extend the original fruit fly optimization algorithm (FOA) framework, improving the FOA's randomness and search power. The effective intrinsic mode functions (IMFs) are determined by searching for the optimal combination values of the key parameters of the variational mode decomposition (VMD) with the improved chaotic FOA (ICFOA). The kurtosis index is then used to select the IMFs that are most relevant to the fault characteristics information. Finally, the sensitive components are analyzed to identify multiple early fault characteristics and determine detailed information about the faults. Moreover, the approach is evaluated by a simulation signal and a measured signal. The comprehensive evaluation indicates that the approach has clear advantages over other excellent methods in extracting the incipient fault feature information of the equipment and has great potential for application in engineering.

**Keywords:** incipient fault detection; feature extraction; complete Teager operator; variational mode decomposition; intrinsic mode functions; composite chaotic mapping; fruit fly optimization algorithm

## 1. Introduction

Large rotating machines with a low speed and heavy load, such as blast furnace rotary distributors, conticaster steel ladle revolving tables, converter rotary support mechanisms, and other metallurgical equipment used by large metallurgical enterprises, are a special type of mechanical equipment in rotary machinery. Their distinctive features are their complex transmission structure, large load-carrying capacity, and low operating speed. Once a fault occurs, it requires a long maintenance period and may cause personal safety problems and huge economic losses. Thus, an early fault detection method is essential to avoid unexpected accidents. However, the early fault signals of such devices have low energy and a long duration and are mixed with strong background noise. Sparsity and weakness are the significant characteristics of the corresponding fault representation. Therefore, effective early feature information extraction is nontrivial due to compound interference.

The collected signal of rotating equipment has distinct non-linear and non-stationary features [1]. Currently, modern signal analysis technologies are mainly used for time-domain and frequency-domain processing of mechanical fault vibrational signals and to extract the fault signal features hidden in complex signals [2]. The commonly used deterministic methods are usually related to signal time-frequency analysis. STFT is

an efficient time-frequency analysis method, but its resolution of time and frequency is fixed [3]. The wavelet transform (WT) is suitable for frequency localization analysis but is limited by choosing the appropriate wavelet basis function [4]. The S transform has excellent adaptive time-frequency analysis capability, but it is affected by the degree of identification [5,6]. The Wigner–Ville distribution method represents the instantaneous time-frequency relationship of the signal, but it is constrained by the resolution in both the time and frequency domains [7,8]. The high-order statistics method effectively eliminates the influence of symmetric distribution noise but tends to cause interference with the higher-order spectrum [9]. Empirical mode decomposition (EMD) suffers from fundamental problems such as mode mixing, fitting overshoot, and end effect, which limit its application in engineering to a great extent [10–13]. Local mean decomposition (LMD) trends produce signal mutations during demodulation [14,15]. Intrinsic time-scale decomposition (ITD) has shortcomings such as waveform distortion and false components [16,17]. Scholars have made many improvements to the above methods; however, limited by the theoretical framework, these improvements make it difficult to eliminate their own problems.

Fortunately, an effective non-recursive method, termed VMD, was proposed in 2014 [18]. In contrast to recursive mode decomposition methods, VMD can decompose a signal into intrinsic mode functions with an estimated center frequency and limited bandwidth. Thus, the problem of solving mode bandwidth is transformed into a constrained optimization problem, and each mode is solved. Moreover, it effectively overcomes the defects of EMD, LMD, and ITD, and it has been widely studied and applied in electromechanical equipment fault diagnosis [19–22]. However, the critical parameters of VMD, such as the total mode number $K$ and mode frequency bandwidth control parameter (quadratic penalty term $\alpha$), must be set in advance, which is highly empirical and blind. Therefore, VMD is not a model-adaptive method in the actual application process. Several recent studies have focused on VMD parameter setting. Based on personal experience and convenience, Zhang et al. solved the difficult problem of milling chatter detection by combining VMD with energy entropy [23]. In [24], a kind of FFT method was introduced to determine the optimal parameters of VMD, but FFT has essential defects in processing nonlinear and non-stationary information [25]. Long et al. [26] used a PSO algorithm to realize the optimization of adaptive values $K$ and $\alpha$. Wang et al. [27] proposed a PSO-based VMD approach for the self-selection of the intrinsic parameters and set the minimum mean envelope entropy as the optimality function. The method improves the function of the original VMD and makes it self-adaptive. To avoid the PSO algorithm falling into the local optimal solution, some scholars have proposed corresponding optimized algorithms, referred to as quantum particle swarm optimization (QPSO) [28]. As the classical improved algorithm of standard PSO, QPSO can enhance the global searching ability significantly. However, the tendency toward precocity still occurs.

With the vigorous development of intelligent optimization technology, Pan et al. [29] introduced an original bionic intelligent optimization approach inspired by fruit fly foraging behavior in 2012, called the fruit fly optimization algorithm (FOA). Recently, it has been extensively studied in many areas of engineering optimization [30]. Compared with the PSO algorithm, the FOA algorithm is easy to program and implement with few control parameters and is convenient for embedding specific search links for the problem [31]. However, FOA is also a heuristic optimization algorithm, and there is still the challenge of falling into the local optimal solution. Ref. [32] proposed a three-dimensional underwater sensor network coverage enhancement optimization algorithm based on an improved fruit fly optimization algorithm (UFOA). Zhang et al. [33] developed a method of adding mutation strategies to a chaotic FOA algorithm, which improves the premature problem to some extent. However, the proposed mutation strategy does not have a strict mathematical definition or detailed comparative analysis.

The weak nature of the vibration impulse represents an early defect of the device, making direct analysis in both the time and frequency domains difficult [34]. The Teager operator improves the shock energy in the vibrational signal and is characterized by

excellent real-time performance, high resolution, and low computational complexity. Fault feature identification requires that the input data be stable and can reliably represent this type of signal. Consequently, screening of sensitive signals is particularly important. The minimum permutation entropy criterion was explored to select the basis function to compute the statistical properties of the signal [35]. The correlation coefficient and variance contribution rate between the reconstructed signal and the original signal are used to determine the optimal choice from the obtained stability criterion [36]. Based on the aforementioned introduction, this work introduces an adaptive CTeager-VMD for early fault feature extraction methods. The main highlights are summarized below: (1) The original vibrational signal is enhanced using the complete Teager operator. (2) A three-dimensional logistic–sine composite chaotic map (LSCCM) is constructed to promote the global search capability and convergence rate of the FOA. (3) The improved FOA is used to seek the optimal parameters $K$ and $\alpha$ of the VMD, and it makes the VMD adaptive to achieve the best processing power. (4) The enhanced vibrational signal is decomposed into several IMFs using the optimized VMD. According to the mean kurtosis criterion, sensitive IMFs with impact components are selected and used to reconstruct a fresh signal. Next, the envelope spectrum of the fresh signal is calculated, and the characteristic information of the incipient fault is obtained. Finally, the proposed method is evaluated using synthetic and measured signals and compared with other excellent methods. The proposed CTeager-VMD method is able to unambiguously identify the characteristic frequency of the micro-impact. The evaluation and comparison results sufficiently verify the superiority and stability of the proposed method.

The subsequent sections of this paper are organized as follows. Section 2 introduces the proposed adaptive VMD method in detail. In Section 3, the performances are compared by simulated experiments. In Section 4, the performance of the proposed method is verified and evaluated. Finally, the conclusions are summarized in Section 5.

## 2. Proposed Methods

### 2.1. Complete Teager Operator

The linear harmonic oscillator with no decaying free oscillations can be defined as follows:

$$m\ddot{y} + p\dot{y} = 0. \tag{1}$$

Its general solution can be described as $y(t) = Acos(\omega * t + \theta)$. The trend change of the narrowband signal can be analyzed and tracked by simple mathematics. The discrete expression of the energy track operator $\psi$ is defined as:

$$\psi_c[y(n)] = [y(n)]^2 - y(n-1)y(n+1) \tag{2}$$

$\psi_c[y(n)] = A^2\sin^2\omega$ can be obtained by substituting $y(n)$ into Equation (2) and it can localize the change in impact energy. The complete Teager energy operator (CTEO) is defined as $T$:

$$T[y(n)] = \frac{\psi_c[y(n)]}{\frac{\sin^2(\omega)}{\omega^2}} = \frac{\psi_c[y(n)]}{sinc^2(\omega)} = A^2 \cdot \omega^2 \tag{3}$$

Obviously, $T$ is the square product of the instantaneous amplitude of the vibration and its instantaneous frequency. Given that $sinc^2(\omega) < 1(\omega \neq 0)$, so $T$ is larger than the original Teager $\psi$. In contrast to the traditional definition of energy, it increases the product of the frequency square. Because of the high vibration frequency of the transient component, the modulation results of CTEO have the ability to effectively characterize the transient impact component. It should be noted that the energy operator used in all subsequent experiments is CTEO.

$$\left.\begin{array}{l} y(t) = y_1(t) + y_n(t) \\ y_1(t) = \sum_j A_j p(t - jT - \tau_j) \\ p(t) = e^{-Ct}\sin(2\pi f_n t) \\ A_j = 1 + A_i \sin(2\pi f_r t) \end{array}\right\} \tag{4}$$

Equation (4) describes the simulation bearing fault-induced signal. Here, $A_i$ denotes the initial amplitude. $A_j$ represents the signal modulation amplitude. $\tau_j$ represents a slight time shift for each period. $C$ is the damping coefficient of signal oscillation. $f_n$ denotes the resonance frequency and $f_r$ is the rotation frequency. $T$ stands for cycle period. By choosing appropriate values of the parameters, it is possible to simulate the bearing fault signal at low speed. In this test, the detail parameters are selected as $C = 750$, $f_r = 1$ Hz, $f_n = 3000$ Hz, and $A_0 = 0.5$. Moreover, the signal sampling rate $f_s$ is 12 kHz and the acquired data points are $L = 12{,}000$. The time-domain waveform of simulated fault signals is shown in Figure 1.

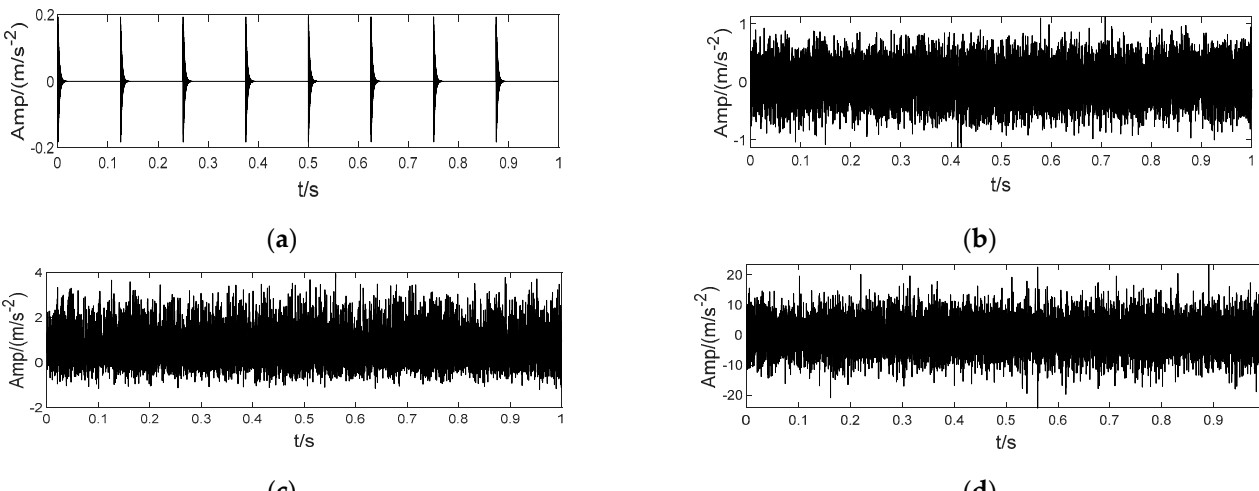

**Figure 1.** SNR enhancement experiment of Teager energy operator. (**a**) Pure shock signal, (**b**) Impulse signal with white noise, (**c**) Original TEO signal, and (**d**) Complete TEO signal.

Figure 1a shows the simulation bearing a fault-induced signal. The repetition period is 20 Hz, maximum amplitude is 0.5 V, sampling frequency is 12 kHz, and the data points are 24,000. Figure 1b shows the impulse signal with white noise; its SNR = 10.1260 dB. Figure 1c shows the signal enhanced by the TEO with SNR = −16.3939 dB. The experimental results show that the TEO can enhance the impulse magnitude and effectively improve the SNR of the impact component in the signal.

### 2.2. Theory of VMD

As a novel signal variational modulation method, the decomposition process of VMD is essentially an iterative solution process of variational problems. From the process, it can be divided into the construction and solution of the variational model.

### 2.2.1. Model Framework

In the VMD process, the input signal is estimated to consist of components with different central frequencies and limited bandwidth [18]. Simply, The sum of the components obtained by the constraint is equal to the original signal.

The unconstrained variational optimization procedure is obtained by introducing Lagrange multipliers $\lambda$ and penalty terms $\alpha$:

$$
\begin{aligned}
L(\{u_k\},\{\omega_k\},\lambda) := &\ \alpha\sum_k \left\|\partial t[F(t)]\right\|_2^2 \\
&+ \left\|f(t) - \sum_k u_k(t)\right\|_2^2 + \left\langle \lambda(t), f(t) - \sum_k u_k(t) \right\rangle
\end{aligned}
\tag{5}
$$

where $\{u_k\} = \{u_1, u_2, \times \times \times, u_K\}$ and $\{\omega_k\} = \{\omega_1, \omega_2, \times \times \times, \omega_K\}$ represent the mode components and center frequencies, respectively. Hence, the saddle point of Equation (5) is obtained by updating $\left\{u_k^{n+1}\right\}, \left\{\omega_k^{n+1}\right\}$, and $\lambda^{n+1}$ with the alternative direction method of multipliers (ADMM). Then, the process obtains $K$-independent representation items $\{u_k\}$.

2.2.2. Performance Analysis of VMD

From the above description, it can be seen that the sub-mode number $K$ and the quadratic term $\alpha$ are closely related to the effectiveness of VMD. Presetting too small $K$ will lead to signal under-decomposition and result in mode aliasing. However, excessive $K$ value can easily lead to modal copying. The parameter $\alpha$ determines the bandwidth of the modes. If this parameter is set too low, the limited bandwidth of each mode is too large, which leads to mode aliasing and introduces noise. Conversely, excessive parameter $\alpha$ will result in the absence of some valid information.

To illustrate the effect of inside parameters on VMD performance, the signal in Figure 1b is decomposed using different $[K, \alpha]$ combinations as shown in Figure 2. Therefore, the experiments show that the $[K, \alpha]$ has a great influence on the performance of VMD. The identification characteristic of the VMD is critically affected by the combination $[K, \alpha]$.

*2.3. Improved Chaotic FOA*

The FOA is a novel swarm intelligence optimization method derived from the principles of Drosophila foraging behavioral ecology [29]. FOA emulates Drosophila with its keen olfactory and visual foraging processes to achieve population-optimal search in the solution space. For improved FOA, we propose a hybrid framework considering two different operators and LSCCM.

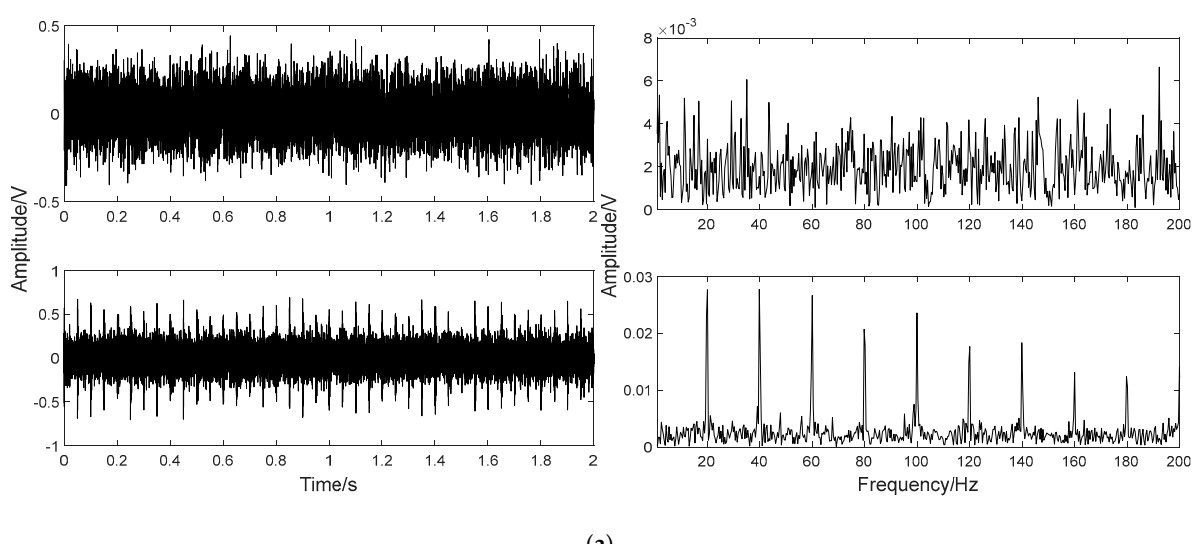

(a)

**Figure 2.** *Cont.*

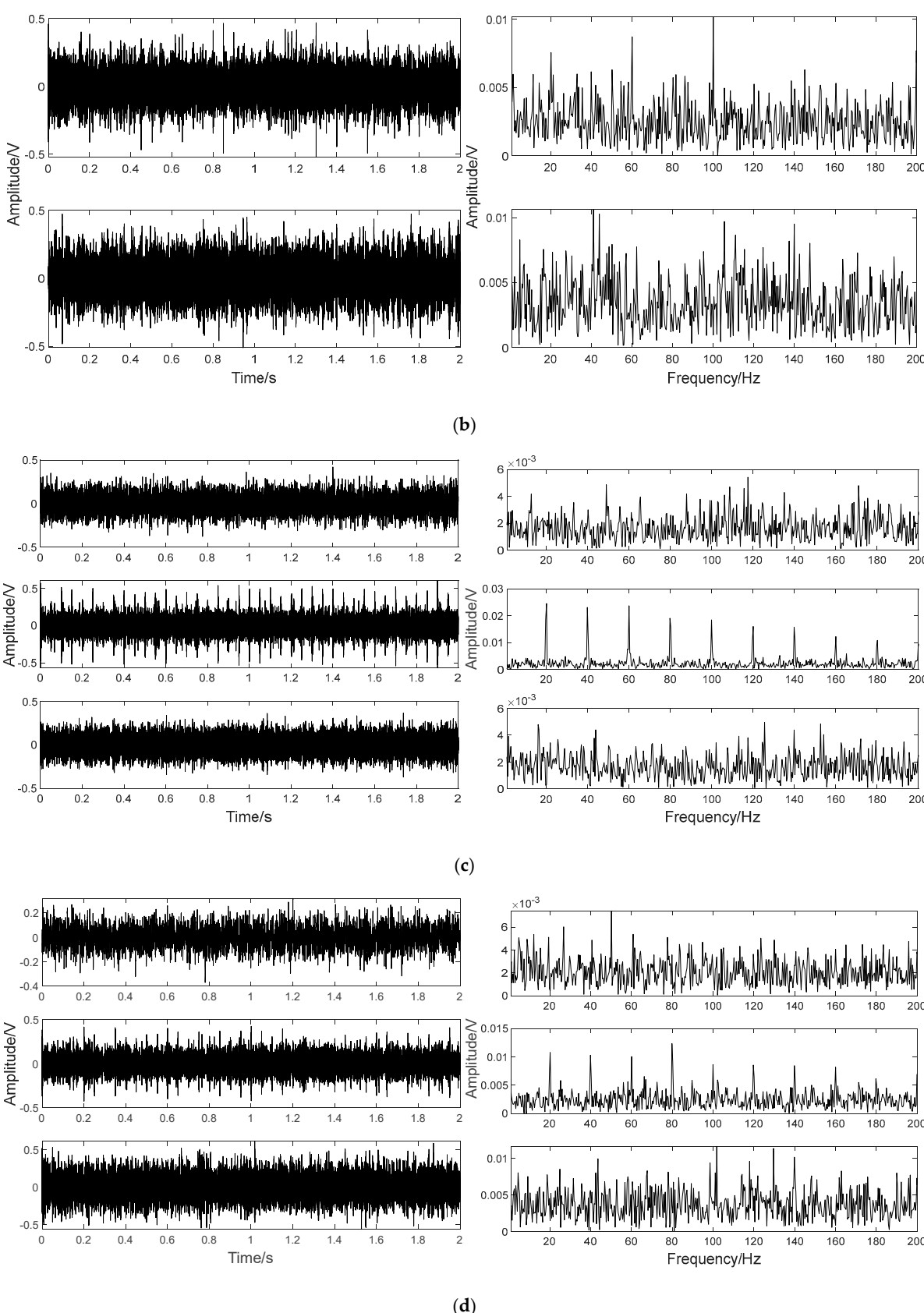

**Figure 2.** Decomposition results with different [*K*, *α*] values. (**a**) [2, 200], (**b**) [2, 1000], (**c**) [3, 200], and (**d**) [3, 1000].

### 2.3.1. A 3D Extension of FOA

When flies are far from a food source, they rely on olfaction to locate targets. Within sight, flies use vision to approach food. Inspired by the original FOA operating principle, this paper introduces an implemented model in which two different operators and a chaotic algorithm are combined.

Smell operator: When looking for food, the fruit fly constantly adjusts its flight position through the sense of smell. When it is close to the food source, it relies less on its sense of smell.

Vision operator: As fruit flies approach a food source, they rely primarily on vision to identify nearby flies and food sources. If the flies can identify a food source through their own vision, they head directly toward it. Otherwise, they follow other flies that have already found a food source.

First, the initial location of the fly is randomly determined. In multidimensional flying space, the location of the individual is updated in every iteration. The initial position is defined as follows:

$$\begin{cases} X_{axis} = Value \times rand(\cdot) \\ Y_{axis} = Value \times rand(\cdot) \\ Z_{axis} = Value \times rand(\cdot) \end{cases} \tag{6}$$

where $Z_{axis}$ is the extension of the original FOA from two to three dimensions, as shown in Figure 3. $Value \in [-bound, bound]$ represents the position boundary. Then, the current positions of all individuals in the population are updated by Equation (7).

$$\begin{cases} X_i = X_{axis}\mathrm{e}^{-R \times I_c} + a\widetilde{X}_i - b \\ Y_i = Y_{axis}\mathrm{e}^{-R \times I_c} + a\widetilde{Y}_i - b \\ Z_i = Z_{axis}\mathrm{e}^{-R \times I_c} + a\widetilde{Z}_i - b \end{cases} \tag{7}$$

where $X_i$, $Y_i$, and $Z_i$ indicate the three-dimensional positions corresponding to the $i-$th fruit fly ($i$ is an integer between 1 and $N$). $R \in [0, 1]$ is smell operator. $I_c$ denotes the current iteration, and the parameters $a$ and $b$ are variables in real.

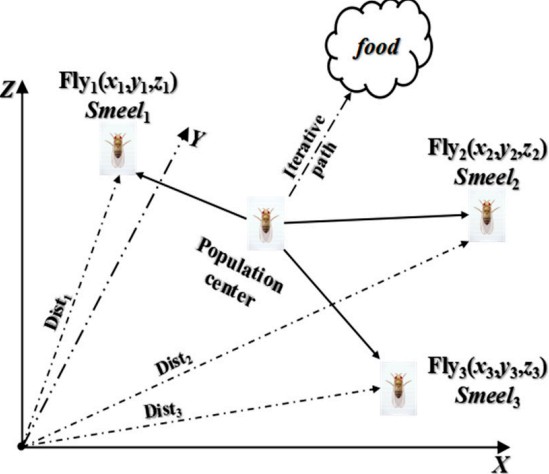

**Figure 3.** Three-dimensional spatial foraging map of fruit flies.

### 2.3.2. Analysis of 3D Logistic-Sine Composite Chaotic Map

Chaotic maps are mathematical relations that give rise to certain chaotic evolution over time [37]. Chaos theory is applicable to dynamic systems sensitive to initial conditions. The evolution of the chaotic function constitutes a sequence of disordered numbers. The chaotic features of the chaotic map are used to initialize the population of fruit flies, ensuring path diversity among population individuals. Currently, a variety of chaotic map functions are in use. Among them, logistic and sinusoidal maps are commonly used, which are characterized by simple models and preeminent chaos [38]. According to the

search mechanism of the FOA, a 3D composite map is constructed by fusing the logistic and sinusoidal maps, and it has much better chaotic characteristics. The 3D-LSCCM is defined as follows:

$$
\begin{cases}
\widetilde{X}_{i+1} = \sin((4\theta \times \widetilde{X}_i(1 - \widetilde{X}_i) + (1 - \theta)\sin(\pi \times \widetilde{Y}_i))\pi) \\
\widetilde{Y}_{i+1} = \sin((4\theta \times \widetilde{Y}_i(1 - \widetilde{Y}_i) + (1 - \theta)\sin(\pi \times \widetilde{X}_{i+1}))\pi) \\
\widetilde{Z}_{i+1} = \sin((4\theta \times \widetilde{Z}_i(1 - \widetilde{Z}_i) + (1 - \theta)\sin(\pi \times \widetilde{Y}_{i+1}))\pi)
\end{cases}
\tag{8}
$$

Equation (8) represents the three-dimensional extension of LSCCM, where $k \in [0, 1]$ is the chaos regulating parameter. $x_n$, $y_n$, and $z_n$ denote variables for composite mapping and they all take values between 0 and 1. Three-dimensional LSCCM expands the search space of the original FOA from two to three dimensions and optimizes the addressing power of the FOA for high-dimensional spaces. Figure 4a–c shows the Lyapunov exponent and chaotic orbit of three 3D composite maps.

According to iteration, the orientations of all individuals in the population are continuously checked to obtain the nearest position of the population. When the iterations reach the set upper limit, the smell operator stops working and the visual factor starts to play a decisive role.

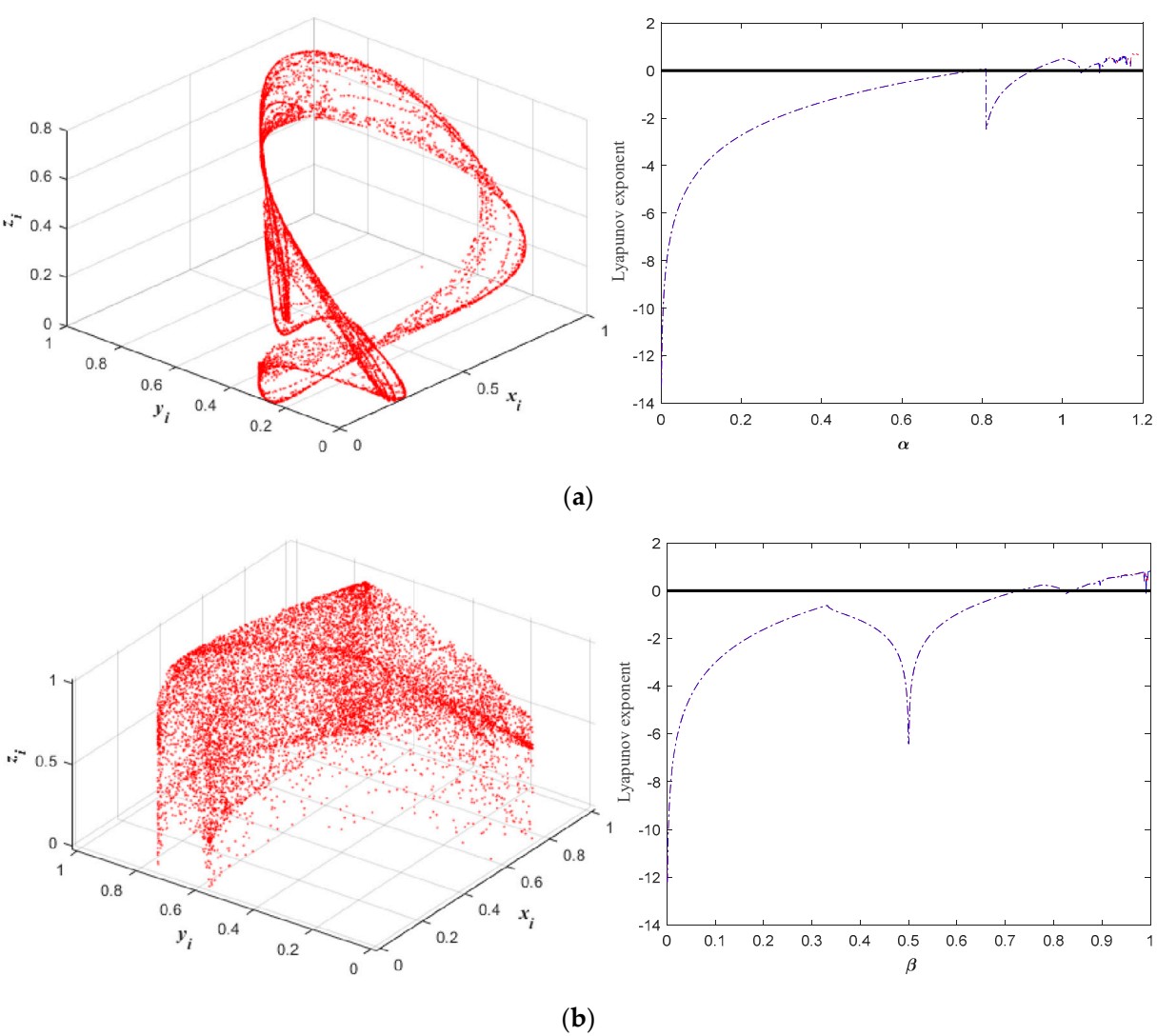

(**a**)

(**b**)

**Figure 4.** *Cont.*

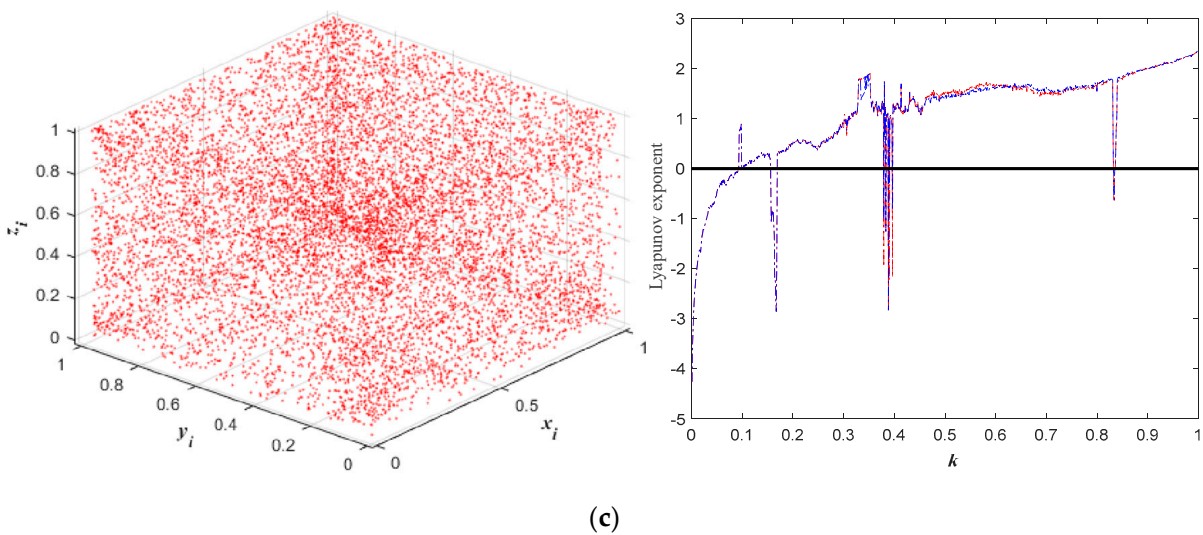

(**c**)

**Figure 4.** Chaotic orbits and LEs of 3D composite mapping. (**a**) 3D LCCM, (**b**) 3D SLCM, and (**c**) 3D LSCCM.

In each iteration of the visual operator decision, the number of individuals will remain half of the current number. The individuals far from the target will be discarded. $S_{centre}$ represents the median of the retained individuals. It is believed to be the next target in the search for food sources. Thus, the positions of fruit flies are updated according to the following equations.

$$S_{centre}^{I_c-1} = \frac{\sum_{i=1}^{N^{I_c-1}} S_i^{I_c-1} F(S_i^{I_c-1})}{N^{I_c-1} \sum_{i=1}^{N^{I_c-1}} F(S_i^{I_c-1})}. \tag{9}$$

$$N^{I_c} = \frac{N^{I_c-1}}{2} \tag{10}$$

$$S_i = S_i^{I_c-1} + rand(S_{center}^{I_c-1} - S_i^{I_c-1}). \tag{11}$$

Similarly, the vision operator stops working when the above cycle reaches the upper limit. Figure 3 illustrates the foraging process of fruit flies. The procedure of the improved chaotic FOA (ICFOA) is illustrated in Appendix A (Algorithm A1).

### 2.3.3. Performance of ICFOA

To evaluate the global optimization performance and convergence rate of the improved chaotic FOA, PSO, QPSO, FOA, CFOA, and ICFOA are applied to test different evaluation indicators. The benchmark functions are as follows:

$$\begin{cases} f_1(x) = \sum_{i=1}^{D} \left(100 * (y_i^2 - y_{i+1})^2 + (1 - y_{i+1}^2)\right) \\ f_2(x) = \sum_{i=1}^{D} \left(y_i^2 - 10 * \cos(2 * \text{pi} * y_i) + 10\right) \\ f_3(x) = \sum_{i=1}^{D} \left(0.5 + \frac{\sin^2 \sqrt{y_i^2 + y_{i+1}^2} - 0.5}{[1.0 + 0.001 * (y_i^2 + y_{i+1}^2)]^2}\right) \end{cases} \tag{12}$$

In all experiments, common parameters such as maximum iteration number, population number were chosen the same for all algorithms. The dimension of the test function is 10, the upper limit of iteration times of all functions and population quantity are set as 200 and 20, respectively. The remaining specifications are set by related literatures. The experimental results are shown in Figure 5.

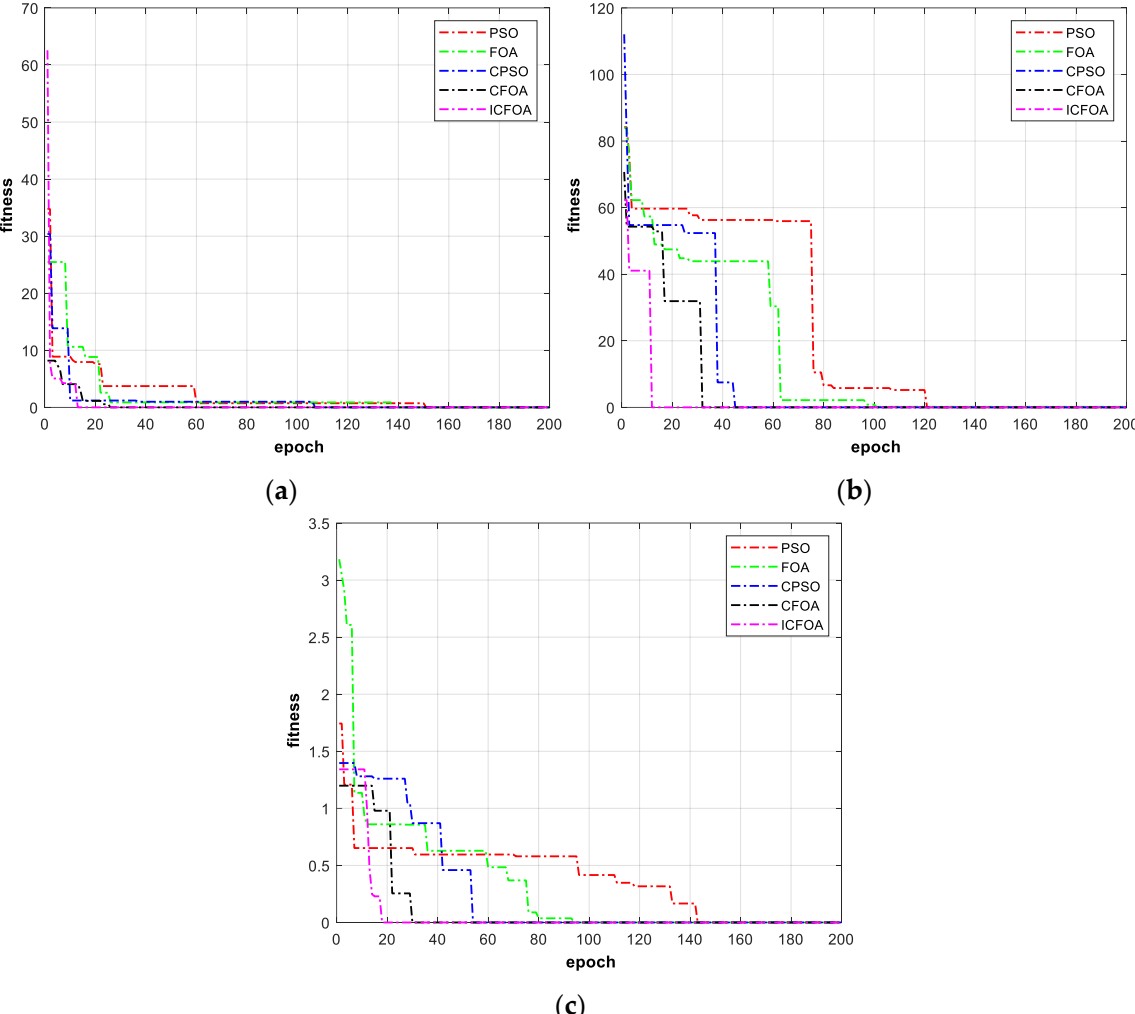

**Figure 5.** Convergence curves of ICFOA and other optimization algorithms. (**a**) Evaluation function $f_1(x)$, (**b**) Evaluation function $f_2(x)$, and (**c**) Evaluation function $f_3(x)$.

For evaluation functions in Equation (12), the proposed ICFOA has the advantage of few iterations and a rapid convergence rate. In addition, in order to verify the reliability and stability of the compared algorithms, all algorithms were run 100 times with different random time periods to find the global optimum solution. The experimental effects are presented in Table 1. Significantly, the proposed ICFOA falls into the local optimum value less than other methods in the process of searching for a global optimal solution 100 times repeatedly. By analyzing the inherent characteristics of the FOA, it can be inferred that the random coordinate induced by the FOA cannot traverse the entire space, and the method may fall into the local optimum. It is worth noting that chaotic maps have the advantages of original value sensitivity and space ergodicity, which can ameliorate such drawbacks to some extent. Therefore, the proposed ICFOA method can effectively escape from the local optimum and has good reliability and stability.

### 2.4. Optimal VMD Algorithm Based on ICFOA

The VMD is not a parameter-adaptive signal processing method, so it is necessary to pre-set relevant parameters in practical application. The operation effect of VMD is significantly affected by the choice of presented group $[K, \alpha]$. In this work, the combined value $[K, \alpha]$ of the VMD was optimized by the ICFOA, and an adaptive VMD algorithm was constructed.

**Table 1.** Statistical results of global optimal solutions on three different benchmark functions.

| Function | Algorithm | Local Optimal Solution | Global Optimal Solution | Iteration Number | Repeat Times |
|---|---|---|---|---|---|
| $f_1(x)$ | PSO | 26 | 74 | | |
| | FOA | 19 | 81 | | |
| | CPSO | 13 | 88 | | |
| | CFOA | 12 | 88 | | |
| | ICFOA | 2 | 98 | | |
| $f_2(x)$ | PSO | 34 | 66 | | |
| | FOA | 32 | 68 | | |
| | CPSO | 26 | 74 | 200 | 100 |
| | CFOA | 22 | 78 | | |
| | ICFOA | 7 | 93 | | |
| $f_3(x)$ | PSO | 28 | 72 | | |
| | FOA | 24 | 76 | | |
| | CPSO | 17 | 83 | | |
| | CFOA | 14 | 86 | | |
| | ICFOA | 4 | 96 | | |

In order to acquire the optimum VMD parameters, in this study, we applied the ICFOA to automatically select the VMD optimization parameters. Equation (13) is used as the objective optimization function of the ICFOA.

$$Fitness_{(ICFOA)} = \frac{\sum\limits_{i=1}^{K-1} MI[u(i), u(i+1)]}{MI[S_{original}, S_{reconstructed}]} \tag{13}$$

where $MI[u(i), u(i+1)]$ represents the mutual information (MI) between neighboring components. A smaller cumulative sum indicates a weaker correlation between IMF components, that is, better orthogonality. $MI[S_{original}, S_{reconstructed}]$ is the mutual information between the raw signal and the reconstructed one [39]. A larger mutual information value indicates a smaller reconstructed error. In addition, MI can measure the nonlinear relationship between two signals. This approach is not restricted to simple linear relations but also can be used to evaluate the non-linear relationship between variables. Therefore, according to the above analysis, as shown in Equation (13), when $\sum_{i=1}^{K-1} MI[u(i), u(i+1)]$ is the minimum and $MI[S_{original}, S_{reconstructed}]$ is the maximum, their ratio is the minimum, that is, the fitness value is the minimum. The analysis indicates that the fault signal is correctly divided into the required components. The information coupling between IMF components is minimal, and the signal reconstruction error is the smallest. Thus, the corresponding optimal parameters $[K, \alpha]$ obtained can allow the algorithm to exert the optimal decomposition performance.

*2.5. Fault Feature Extraction*

The critical component bearings of large, low-speed, and heavily loaded machinery are vulnerable to damage. Therefore, when an early micro-fault occurs in a bearing (i.e., an inner ring defect and an outer ring defect), the energy in the collected vibrational signal representing the shock component of the fault is severely inconspicuous. Effective information is difficult to obtain using common time-frequency analysis methods from raw vibrational signals. Therefore, CTEO was used to enhance the raw vibrational signal in this study. Then, the optimal VMD was used to decompose it to obtain several IMF components, and the kurtosis value of each IMF component was calculated. The mean kurtosis criterion was then used to screen the IMF components, and the selected IMF components were reconstructed into the current signal [40]. Finally, if the bearing of the mechanical equipment fails (such as the inner ring or the outer ring), the vibration signal

collected will contain a specific frequency impact component. The proposed method is then used to decompose and reconstruct the vibration signal. Then, the envelope spectrum of the reconstructed signal is analyzed, and the characteristic frequency of the corresponding fault is extracted from the envelope spectrum. The fault characteristics and extraction flow is demonstrated in Figure 6. Moreover, the specific steps are listed below:

Step 1: The impact component of the fault signal is enhanced by CTEO.

Step 2: The relevant parameters of the 3D-LSCCM are initialized and an appropriate indicator function is designed.

Step 3: The preset values of the fruit fly population are initialized. The binding group $[K, \alpha]$ of the VMD corresponds to the location of the individual fly.

Step 4: Decompose the signal by VMD under different individual positions of flies, and calculate the fitness of each location.

Step 5: The fitness values of the individuals in the population are compared, and the optimal evaluation values of the individuals and the population are updated.

Step 6: Update the position of the individual flies by using Equations (7) and (8).

Step 7: Repeat the iterative procedure of steps 4–6. When the iterative number reaches the maximum set value, the optimal parameters determined by the evaluation function are recorded.

Step 8: Use the obtained parameter combination $[K, \alpha]$ to construct the optimal VMD.

Step 9: Calculate the kurtosis of every independent mode and find the mean of all their kurtosis values. Based on the mean value, IMF components with kurtosis larger than the mean are chosen and reconstructed to obtain fresh representation.

Step 10: The envelope spectrum of the new representation is calculated and analyzed. Afterward, it is matched to the fault feature to determine the corresponding defect type.

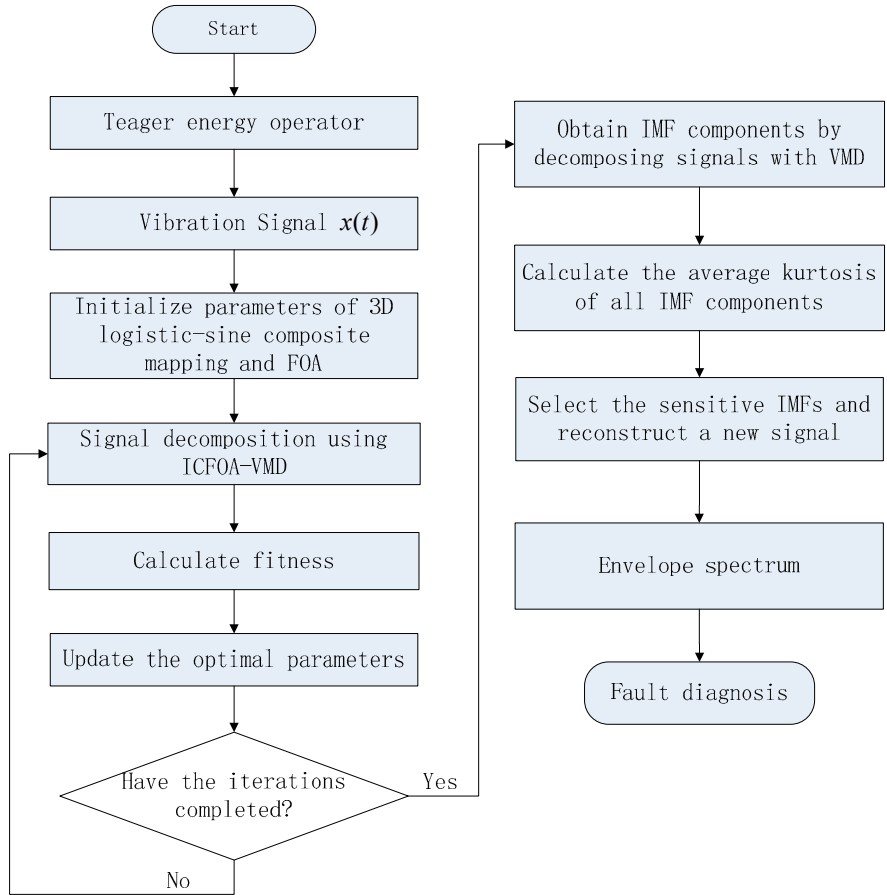

**Figure 6.** Fault diagnosis process using proposed CTeager-VMD.

## 3. Simulated Signal Evaluation

Bearings, as a key part of heavy-load mechanical equipment, are prone to pitting, corrosion, or cracking in long-term operation. When other parts make contact with or impact the defect location, a periodic shock signal is generated. The signals of rotating machinery measured by acceleration sensors are analyzed. When it fails, the fault signal is a sinusoidal signal with exponential decay. The sparsity and weakness are the significant characteristics of early faults in vibration signals. It is particularly susceptible to interference from strong noise or other ineffective signals, and no useful characteristic information reflecting the fault can be identified for fault diagnosis. In order to evaluate the reliability and superiority of our proposed method, a simulation case was used for analysis and verification as follows:

$$y(t) = y_s(t) + y_n(t) \tag{14}$$

where $y_s(t)$ A is a sinusoidal decaying signal with a pulse period of 8 Hz, which is used to simulate the pulse response signal caused by the shock. $y_n(t)$ denotes a white Gaussian noise. The SNR of $y(t)$ is $X_{SNR} = -20$ dB. The simulated fault impulse $y_s(t)$ of the bearing is defined by Equation (4).

The parameters for fault simulation signal are specified as $f_r = 1$ Hz, $f_n = 3000$ Hz, $C = 750$ and $A_0 = 0.2$. Moreover, the fault characteristic frequency $f_{inner} = 1/T = 8$ Hz, sampling frequency $f_s = 12$ kHz and data length L = 12,000.

The original vibration signal and noise-added signal are shown in Figure 7a,b, respectively. When Gaussian white noise was applied to the pure pulses shown in Figure 7a, the simulated impulse signal was completely submerged by the interferences. The construct signal shown in Figure 7b was simulated to evaluate the actual early fault signal of the bearing properly. Then, the mixed signal was evaluated by CEEMD, LMD, VMD, and ITD, separately. The operation parameters of these algorithms were set according to the corresponding literature. Based on these methods, the fault characteristic frequency (theoretical value 8 Hz, actual value $f_1 = 8.057$ Hz) and its harmonic in the frequency domain is shown in Figure 8.

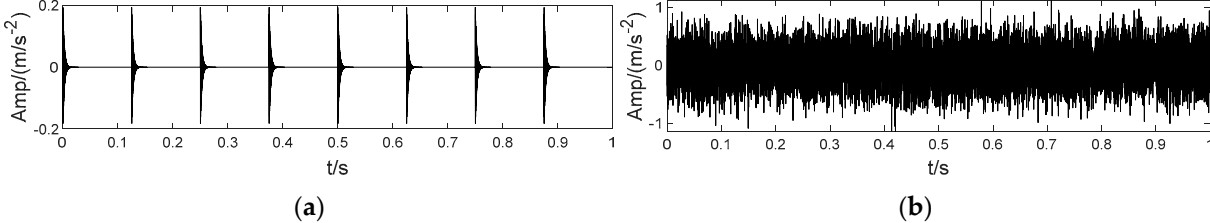

(a)  (b)

**Figure 7.** Simulated signal of bearing failure. (**a**) Raw impulse. (**b**) The impulse with noise.

As shown in Figure 8a–d, the fault representation $f_1$ and its doubling frequency $2f_1$ corresponding to CEMD, LMD, ITD, and VMD are very blurry and low amplitude. Some larger interference frequency components are distributed in the dual frequency band. Compared with other methods, VMD has some advantages in characteristic amplitude, but it is also very weak. Moreover, some signal spectra are completely submerged. The experiments show that when the SNR is low, the above methods lack the ability to extract fault characterization directly.

According to Equation (3), the mixed simulation signal is enhanced by CTEO (shown in Figure 9). The envelope spectrum corresponding to the enhanced signal is analyzed using the same signal processing method as in Figure 8. From the analysis results of the envelope spectrum in Figure 10, it can be seen that the amplitude of the envelope spectrum of fault signal enhanced by the energy operator is obviously improved. In particular, when the VMD method is used, the magnitudes of fundamental and double frequencies of fault

representation are improved noticeably, and the effect is superior to other signal processing methods. However, interference signals in the spectrum are very clear.

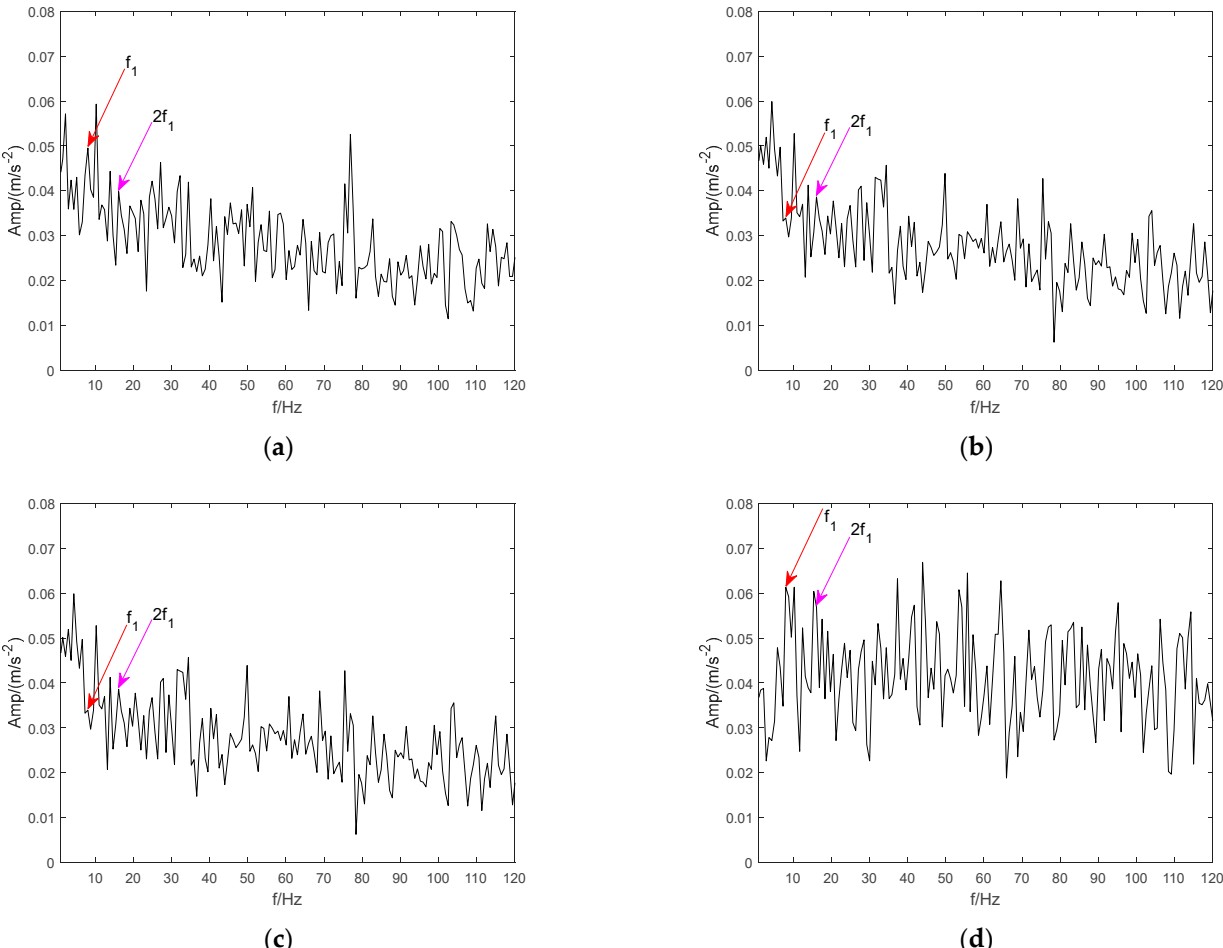

**Figure 8.** The envelope spectrum of bearing fault simulation signals. (**a**) CEEMD. (**b**) LMD. (**c**) ITD. (**d**) VMD.

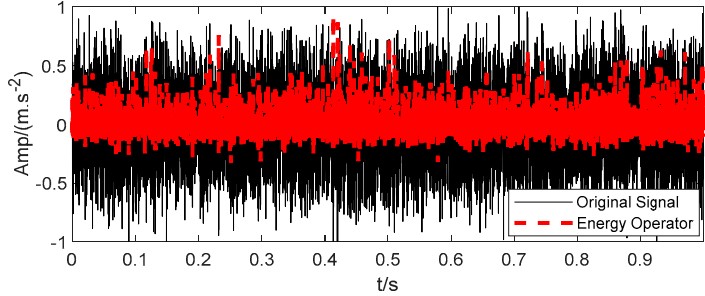

**Figure 9.** Teager energy signal of bearing fault simulation signal.

As shown by the analysis of the principle of VMD, its decomposition performance is significantly affected by parameters $K$ and $\alpha$. However, there is some blindness and contingency in selecting parameters by manual experience, and the performance of VMD is not guaranteed to be optimal. Hence, in order to obtain an optimal VMD and eliminate the invalid signal components in fault signals effectively, the proposed ICFOA is used to optimize the parameters. The process for setting parameters is as follows:

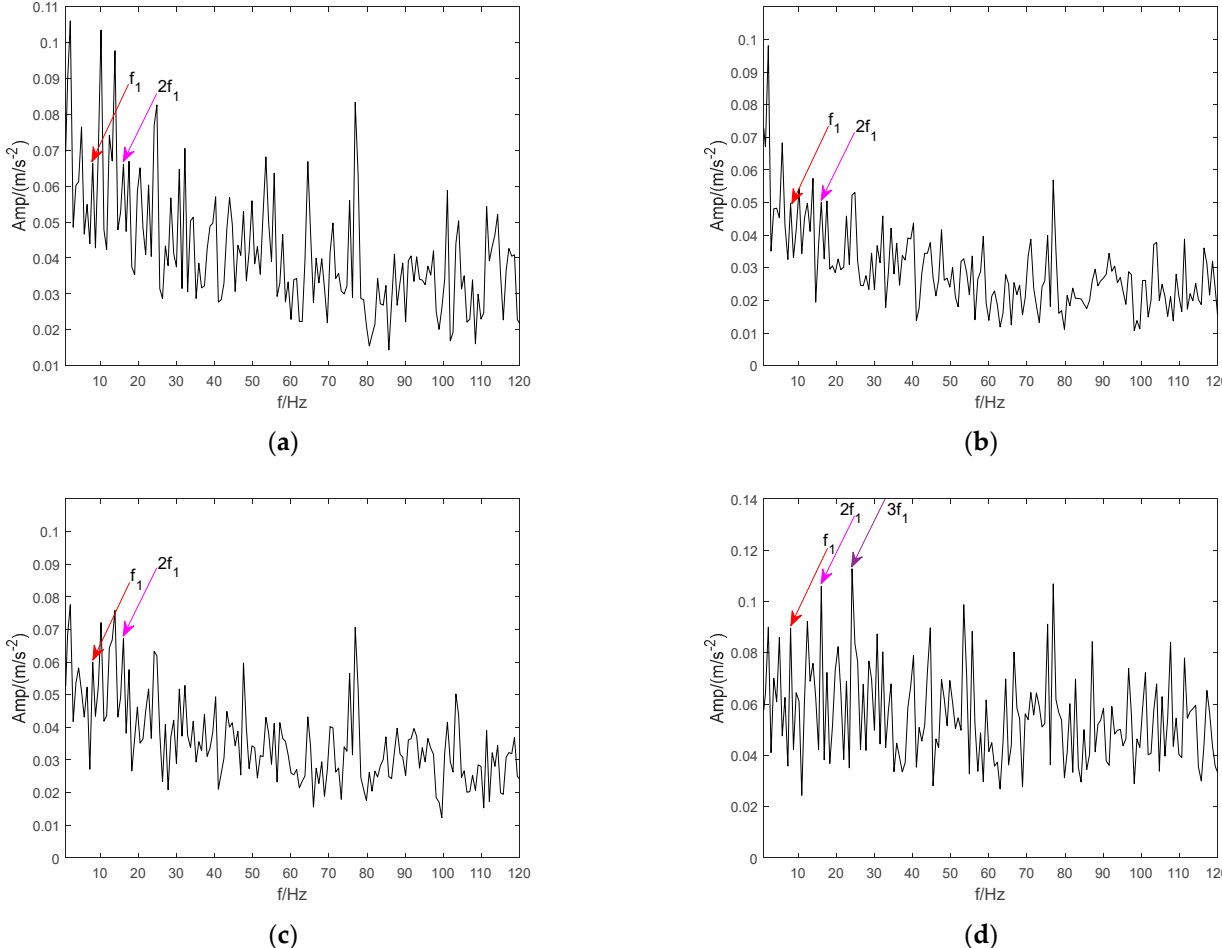

**Figure 10.** Teager energy enhanced envelope spectrum. (**a**) CEEMD, (**b**) LMD, (**c**) ITD, and (**d**) VMD.

Step 1: CTEO is used to improve the impact component in the raw signal.

Step 2: The relevant parameters of the 3D composite chaotic map are initialized with $\theta = 0.99$, $B = [2, 4000]$. The evaluation function of the ICFOA is determined during the optimization procedure.

Step 3: The parameters of the fruit fly population are initialized with $I_{C1max} = 120$, $I_{C2max} = 80$, $Ns = 30$, $D = 10$, and $S = $ 3D-LSCCM.

Step 4: Some combinations $[K, \alpha]$ corresponding to population number are induced by 3D-LSCCM as the reference locations of individual flies. Calculate the standard deviation STD of the simulation signal, set the updating step length of the VMD algorithm to $\tau = 0.003$ based on STD, and set the fault tolerance threshold of convergence to $\varepsilon = 1 \times 10^{-7}$. Then, the 3D-LSCCM is used to update the locations of the individuals in a global search.

Then, the five optimization algorithms in Table 2 are applied to individually select from the optimum group $[K, \alpha]$ for VMD. The effects of the optimization are shown in Figure 11 and Table 2. The proposed ICFOA outperforms other algorithms in terms of iterations and convergence rate. Then, ICFOA is used to search for the optimum group of VMD parameters. Finally, the VMD is reset and used to demodulate the fault signal of the simulated bearing failure. The fundamental frequency (8 Hz) of the fault and its frequency multiplications ($2\times$–$7\times$) can be distinctly detected, as shown in Figure 11b. On comparing the envelope spectrum with Figure 10d, it shows much better clarity and distinguishability on spectrum lines. The compared results indicate that the proposed method is effective in extracting sensitive features.

**Table 2.** Results obtained by different optimization algorithms.

| Parameter | Method | | | | |
|---|---|---|---|---|---|
| | **PSO** | **FOA** | **CPSO** | **CFOA** | **ICFOA** |
| Fitness | 3.318 | 2.055 | 1.656 | 0.7217 | 0.2852 |
| $[K, \alpha]$ | [4, 1714] | [4, 2045] | [5, 2865] | [5, 3065] | [6, 3207] |
| iterations | 85 | 38 | 37 | 44 | 20 |

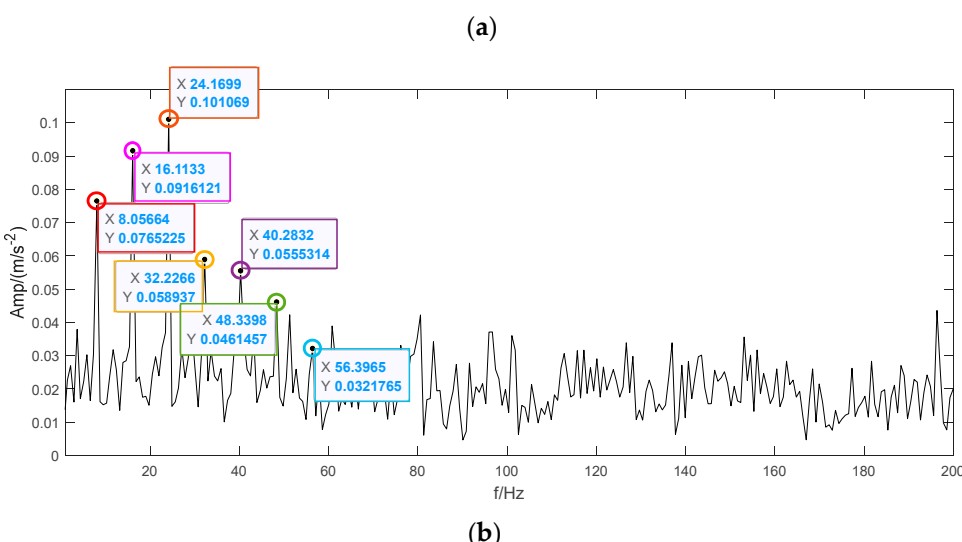

(**a**)

(**b**)

**Figure 11.** Processing results of ICFOA-based CTEO-VMD method. (**a**) Convergence curves for the searching process by five algorithms. (**b**) The envelope spectrum of the reconstructed signal.

## 4. Fault Experiment Analysis

In this section, the feasibility of the introduced method on the actual measurement rig is further verified. As illustrated in Figure 12a, the bench simulates the radial load by applying different radial load forces to the rolling bearing through a hydraulic lever. In addition, the rolling bearing operating speed is adjusted by controlling the motor speed.

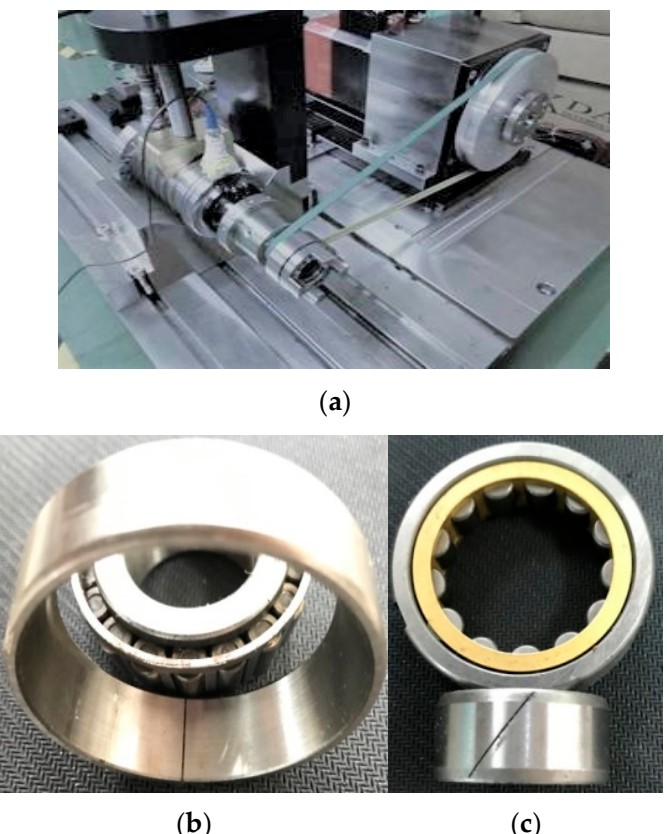

(**a**)

(**b**) (**c**)

**Figure 12.** The experimental rig and defect diagrams. (**a**) Experimental rig. (**b**) Inner ring defect. (**c**) Outer ring defect.

The rolling bearings are shown in Figure 12b,c. Among them, the diameter of the rolling bearing is 39.5 mm, the diameter of the rolling element is 7.5 mm, the number of rolling elements is 12, and the angle of the internal contact surface is 0 degrees. In order to effectively evaluate the operating conditions of low speed and heavy load, the radial load force applied here was set at 5000 N. The rotation speed of the inner ring is 60 r/min, and the vibration signal sampling rate is 10 kHz. Depending on the parameter calculation reference, the theoretical values of outer rings and inner rings can be calculated, and the fault signatures are listed in Table 3. The measured vibration signals of the bearing inner ring fault (BIRF) and bearing outer ring fault (BORF) are shown in Figure 13.

**Table 3.** Rolling bearing fault characteristic frequencies.

| Speed (r/min) | BIRF ($f$/Hz) | BORF ($f$/Hz) |
| --- | --- | --- |
| 60 | 7.14 | 4.86 |

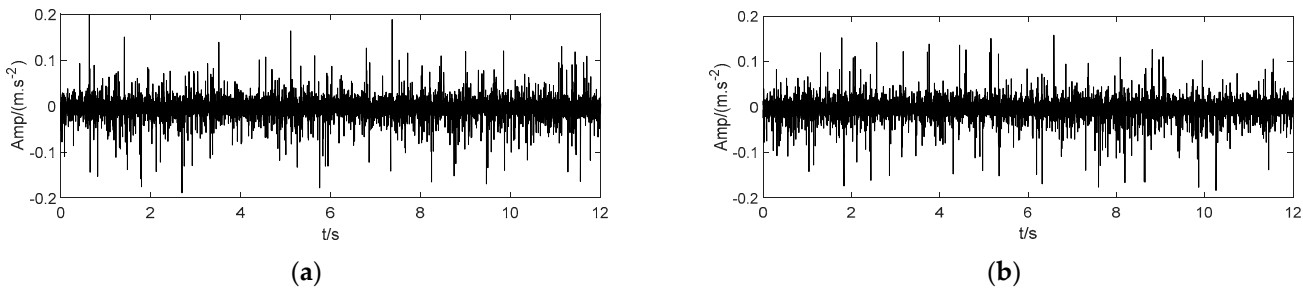

(**a**) (**b**)

**Figure 13.** Vibration signals from rolling bearing fault. (**a**) BIRF. (**b**) BORF.

To evaluate the availability of our method, the signals of identical bearings with inner and outer ring faults were analyzed using standard CEEMD, LMD, ITD, and VMD methods. Figures 14 and 15 show the spectra of BIRF and BORF processed by these methods, respectively. As shown in Figures 14 and 15, the spectral lines of fault characteristics (outer ring $f_{outer}$ and inner ring $f_{inner}$ are 4.88 Hz and 7.17 Hz, respectively) obtained by CEEMD, LMD, ITD, and VMD are fully overwhelmed by additional irrelevant interference spectral lines.

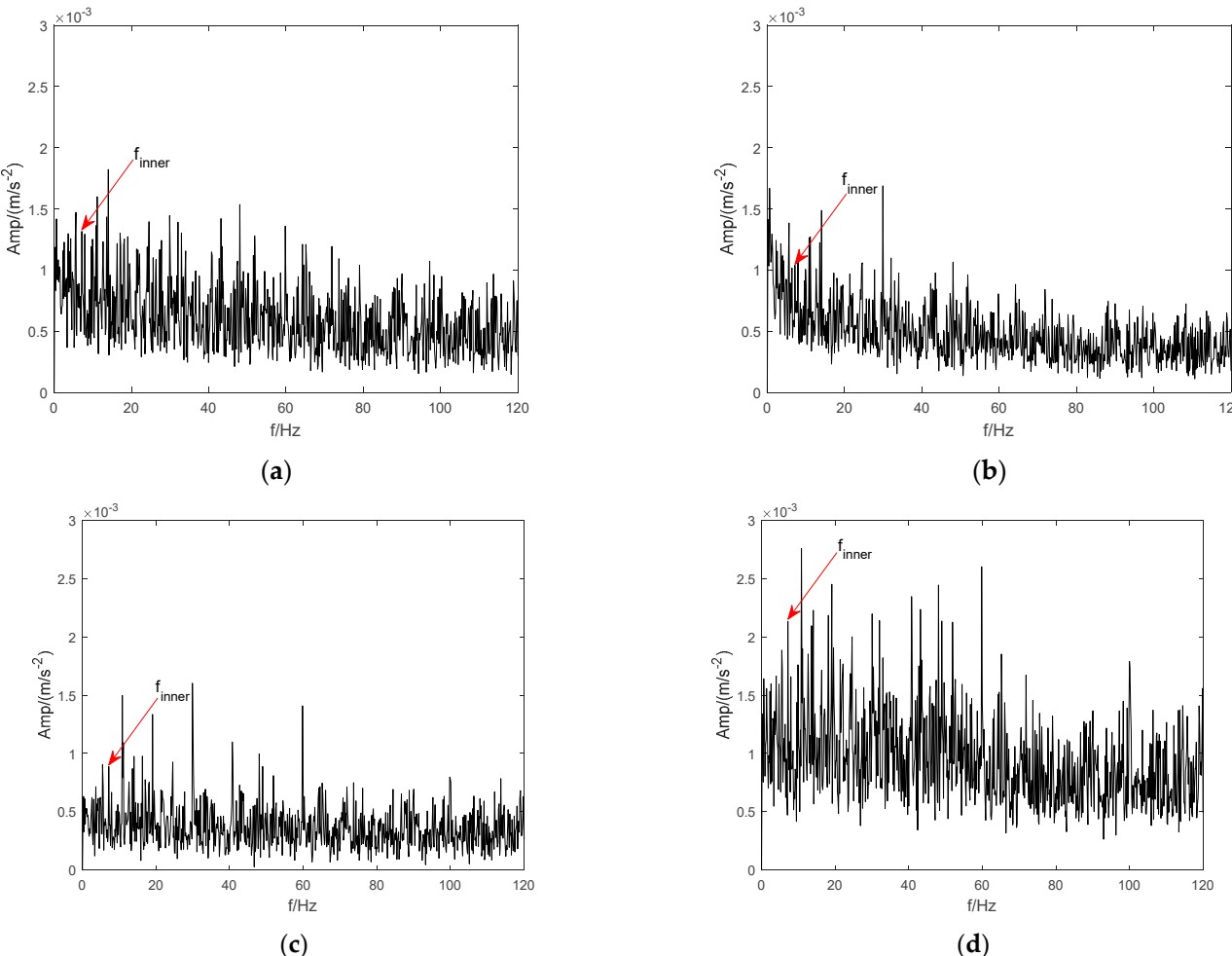

**Figure 14.** Envelope spectrum of BIRF signal. (**a**) CEEMD. (**b**) LMD. (**c**) ITD. (**d**) VMD.

Due to the sparseness and weakness of bearings and early fault signals under low-speed and heavy-load conditions, it is hard to use the above signal processing methods to find useful information for diagnosis. In this study, CTEO was used to preprocess the early fault signal of the bearing to increase the weak impulse component in the raw signal. From Figure 16a,b, it can be observed that the CTEO effectively enhances the impulse components in the fault signal. Then, standard CEEMD, LMD, ITD, and VMD were applied to analyze the enhanced signal, and the envelope spectra are presented in Figures 17 and 18. Among them, Figure 17a–d show the envelope spectra of the inner ring fault signal processed by CEEMD, LMD, ITD, and VMD, respectively. As seen from the spectrum magnitude, the amplitudes of inner ring fault characteristic frequency in Figure 17 are obviously larger than those in Figure 14. Figure 18a–d demonstrate the envelope spectra of the outer ring fault signal with CEEMD, LMD, ITD, and VMD, respectively. The amplitude of each outer ring fault characteristic frequency in Figure 18 is significantly larger than that in Figure 15, too. The experimental results show that using CTEO to preprocess raw faulty signals can effectively enhance the impact component. Furthermore, comparing the envelope spectra

obtained with the classical CEEMD, LMD, and ITD methods, it can be seen that the spectral lines representing the fault feature in the envelope spectra obtained with the VMD method are significantly larger than those obtained with the remaining three methods. Considering the above evaluation and comparison, the VMD method outperforms mechanical vibration signal processing compared to other commonly used methods. Moreover, these comparison results prove that it is a feasible and effective strategy to enhance and preprocess the original fault signal using the energy operator.

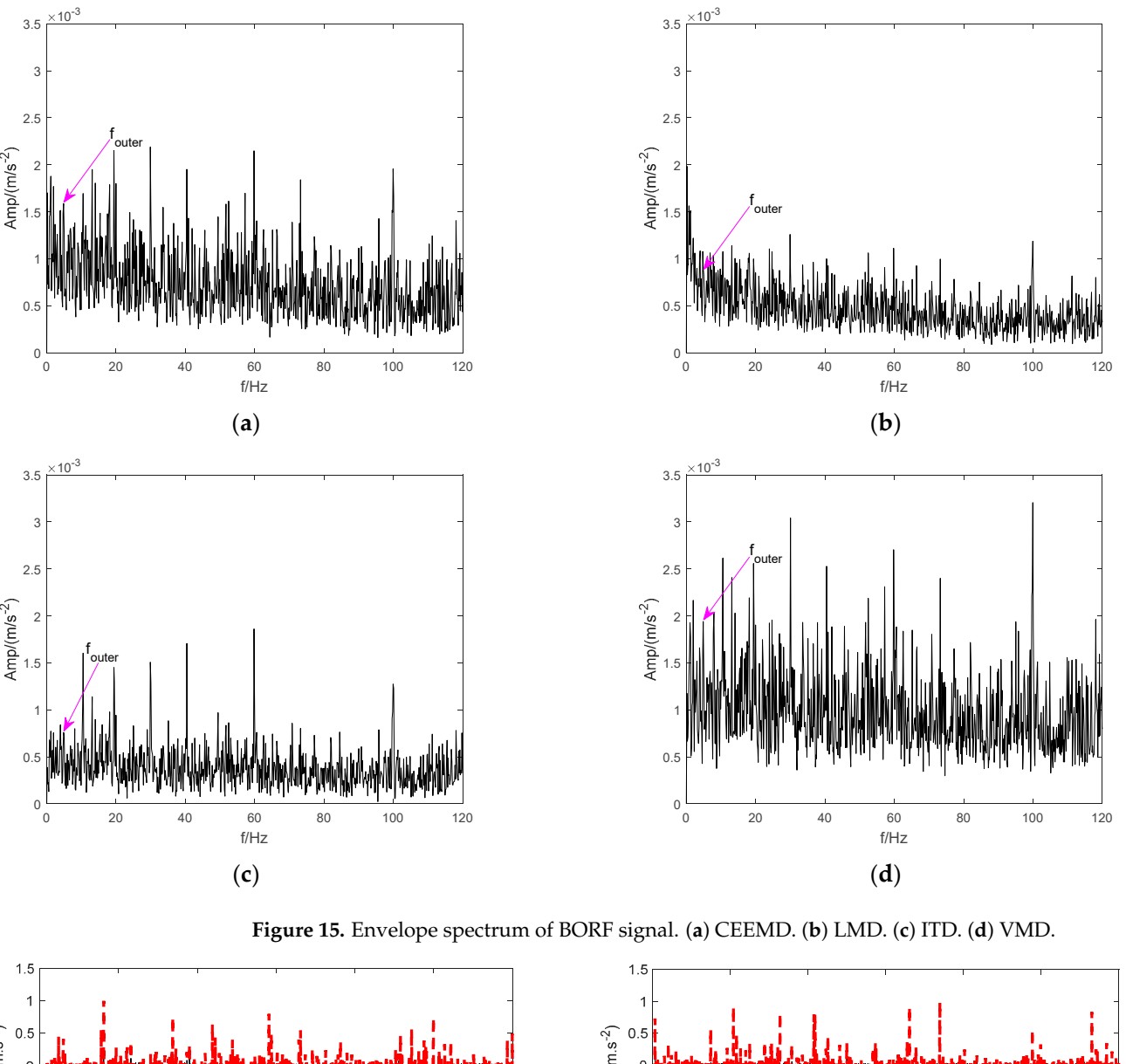

**Figure 15.** Envelope spectrum of BORF signal. (**a**) CEEMD. (**b**) LMD. (**c**) ITD. (**d**) VMD.

**Figure 16.** Bearing fault signal enhanced by CTEO. (**a**) Inner ring. (**b**) Outer ring.

As shown in Figure 17d, by combining CTEO and VMD methods, the fault feature of the BIRF can be effectively identified. However, it is difficult to distinguish the fundamental

frequency from the bi-frequency of this feature, and there is still a large amount of the interference component mixed in the spectrum. Figure 18d shows a similar spectrum distribution to Figure 17d. The reasons for this mainly involve two aspects: (1) The early fault features are severely sparse and faint, nearly overwhelmed by background noise and interference components; (2) The VMD parameters are not optimal. Therefore, in this study, the ICFOA was used to obtain the most suitable combined values $[K, \alpha]$ of VMD, following the same procedure as described in Section 3. In addition, the parameters related to the chaotic mapping and FOA remained unchanged, except that the VMD update step needed to be manually set based on the standard deviation of the raw fault. Firstly, the optimal combination values of VMD searched by the ICFOA were $[K_i, \alpha_i]$ = [13, 3175] and $[K_o, \alpha_o]$ = [11, 2406], and the obtained parameters were saved for each. Then, the optimal approach was used to analyze the BIRF and BORF signals excited by the bearing in Figure 16a and b, respectively. The BIRF signal was decomposed into 13 IMFs (Figure 19a), and the corresponding kurtosis values of each IMF component were 20.1, 22.6, 20.8, 22.7, 22.2, 21.1, 23.1, 16.9, 18.7, 21.6, 24.2, 22.3, and 31.4. The mean kurtosis value is 22.1. The BORF signal was decomposed into 11 IMF components (as shown in Figure 20a), and the corresponding kurtosis values for each IMF component were 14.6, 18.7, 16.4, 20.8, 17.5, 16.2, 18.2, 16.5, 15.2, 17.2, and 24.5, with a mean kurtosis of 17.8. Then, the kurtosis of each IMF component was calculated, the corresponding IMF component with a kurtosis larger than the mean value was selected, and a fresh signal was reconstructed. The reconstructed representations are shown in Figures 19b and 20b. Finally, the envelope spectra of the two reconstructed signals were solved. The final effects are illustrated in Figures 19c and 20c. As shown in Figure 19c, the fault representation (7.17 Hz) and its second harmonics (14.34 Hz) of the BIRF reconstructed signal are effectively extracted, and the interference components are obviously reduced. Similarly, from Figure 20c, the fundamental frequency (4.88 Hz) and double frequency (9.76 Hz) of the reconstructed fault signal can be effectively observed, and the interference spectrum is also significantly reduced. The experimental evaluations demonstrate that the proposed approach can efficiently identify fault features under excessive load and low-speed operation. Additionally, it offers a fruitful means for incipient fault diagnosis of heavy-duty and low-speed rotation equipment.

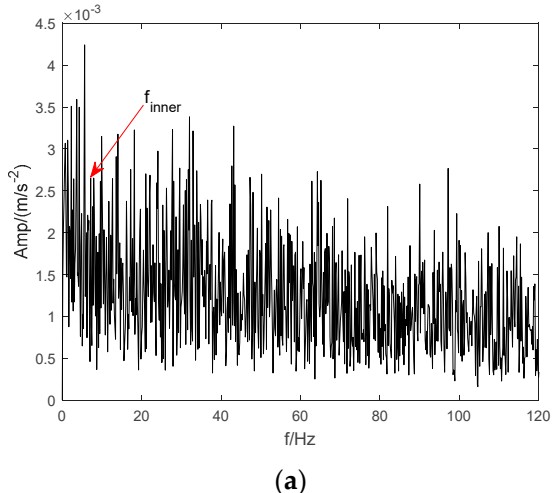

(**a**)

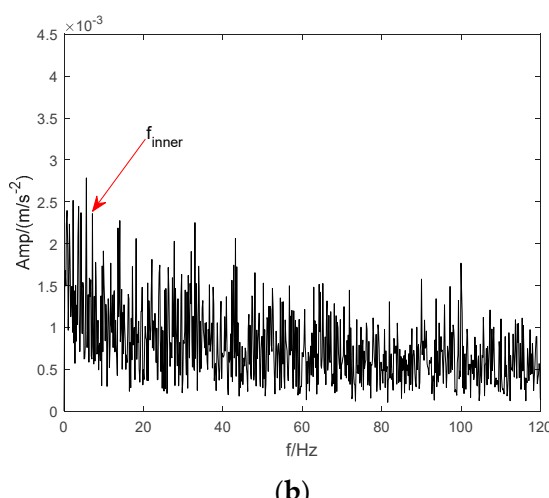

(**b**)

**Figure 17.** *Cont.*

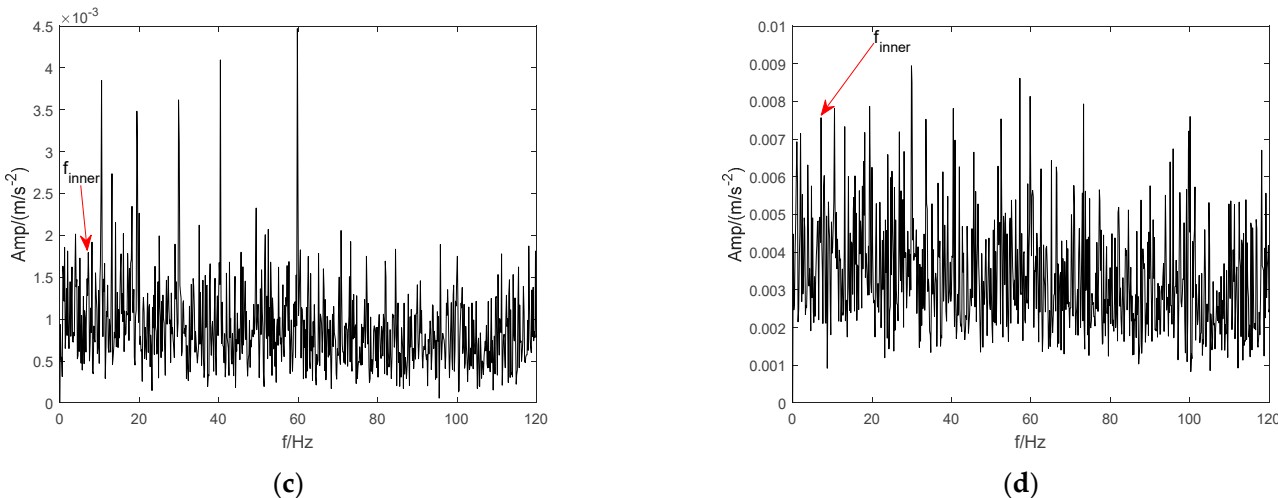

**Figure 17.** The envelope spectrum of BIRF enhanced by CTEO. (**a**) CEEMD. (**b**) LMD. (**c**) ITD. (**d**) VMD.

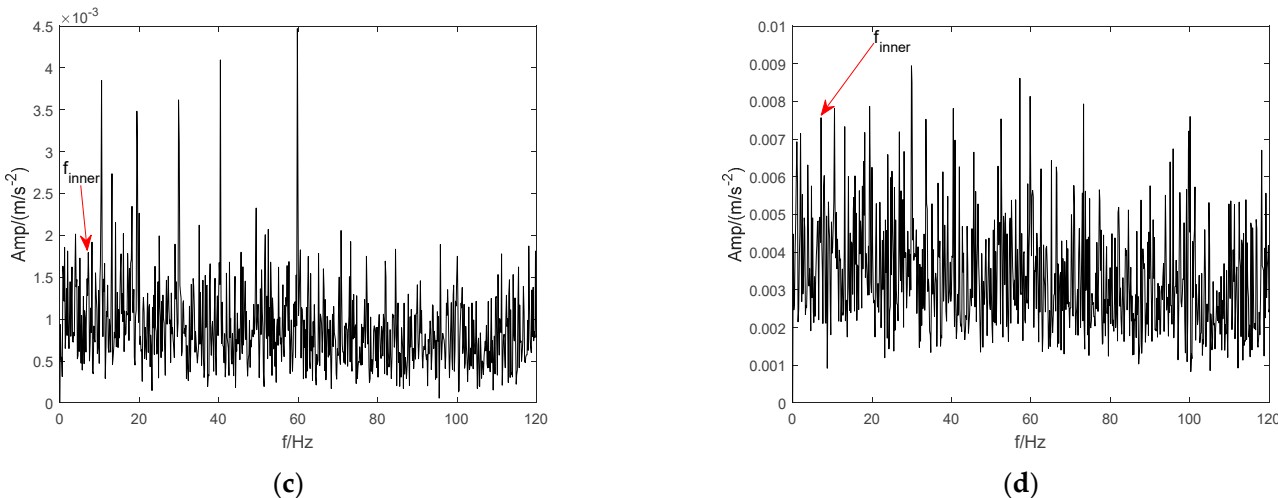

**Figure 18.** The envelope spectrum of BORF enhanced by CTEO. (**a**) CEEMD. (**b**) LMD. (**c**) ITD. (**d**) VMD.

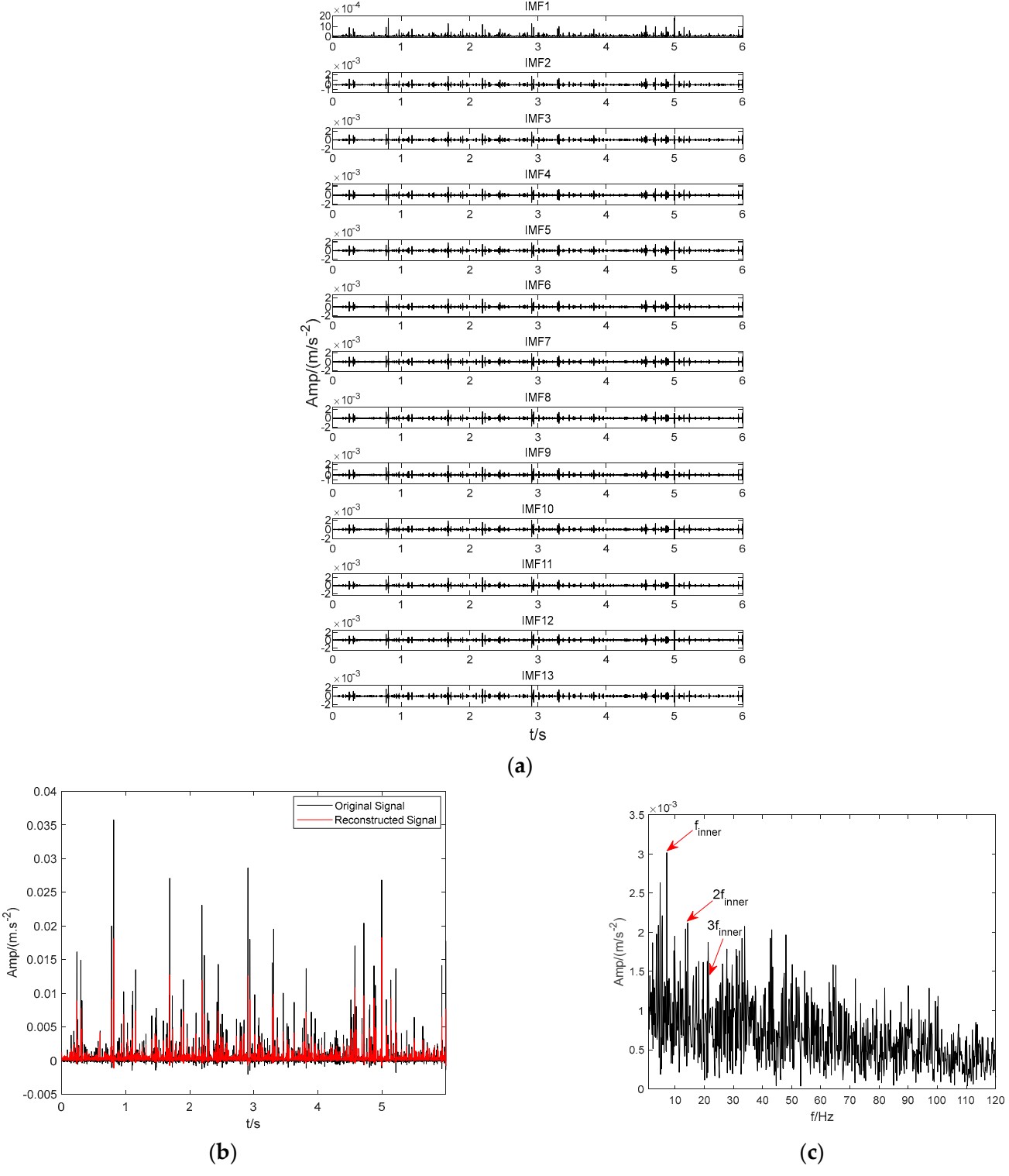

**Figure 19.** Analysis of the obtained modes and reconstructed result based on optimal VMD (BIRF). (**a**) Modes. (**b**) Signal-reconstructed result. (**c**) Spectrum.

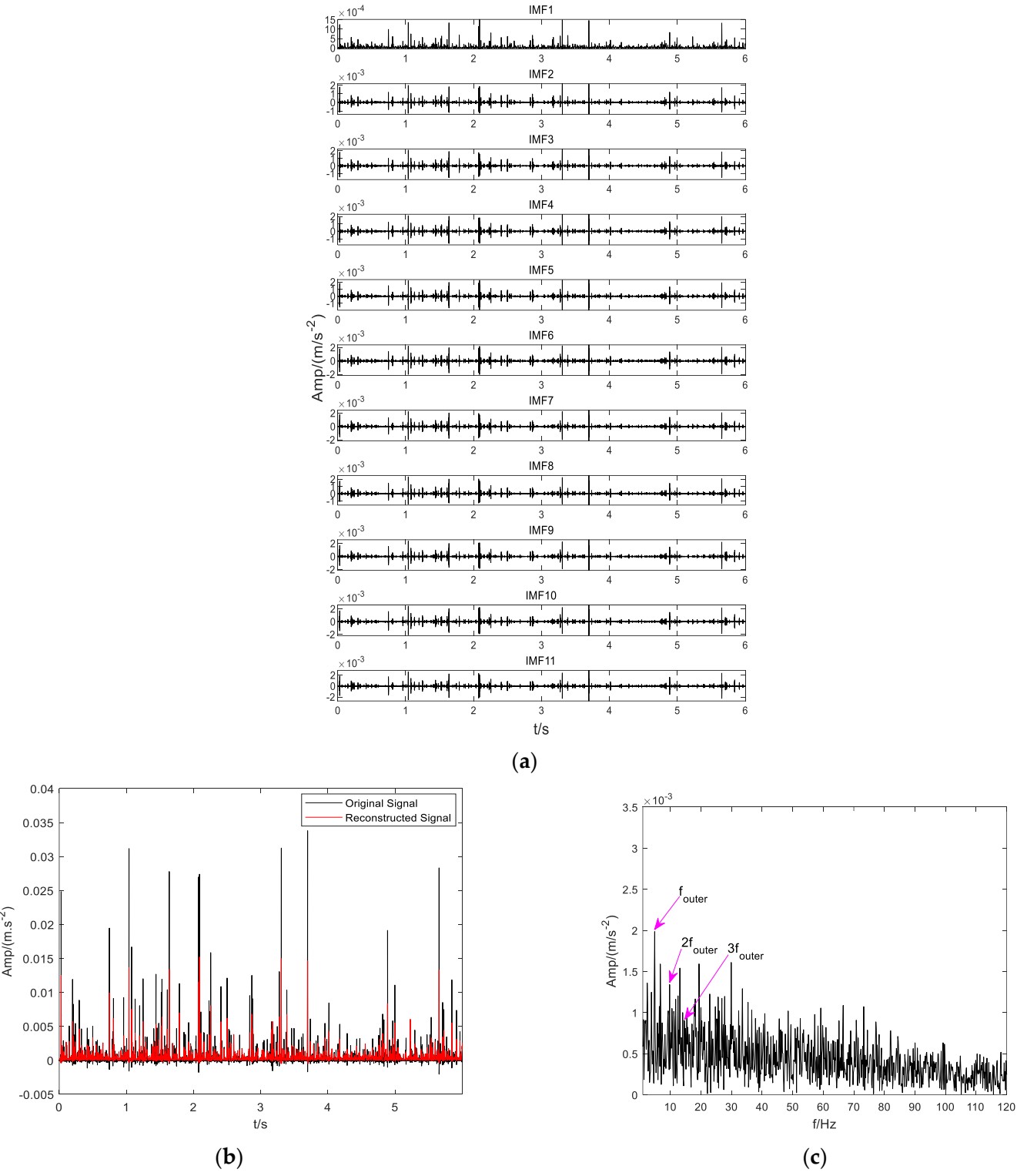

**Figure 20.** Analysis of the obtained modes and reconstructed result based on optimal VMD (BORF). (**a**) Modes. (**b**) Signal-reconstructed result. (**c**) Spectrum.

## 5. Conclusions

In this work, we present a fusion algorithm based on CTEO and parameter-optimized VMD to solve the difficulty of early fault feature extraction for low-speed and heavy-load devices. The application of both simulated and real data shows that the proposed

framework separates the feature components more effectively and preserves effective feature representations more accurately than other signal approaches.

(1) CTEO is an exact value of the conventional TEO, which can further enhance the impulse component in the signal and improve the signal-to-noise ratio of the faulty signal. It should be noted that the early failure signals of large, low-speed, and heavy machinery are very sparse and weak. It is easy to cause VMD to misjudge fault signals as "noise" and fail to decompose them into corresponding IMF components accurately. The use of CTEO to preprocess the raw vibrational signal effectively enhances the signal energy of the impulse component in the signal, which can help VMD properly decompose the impulse signal characterized by the fault into the corresponding IMF components.

(2) Based on the nature of the logistic map and the sine map, we propose a 3D logistic-sine complex chaotic mapping. It extends the two-dimensional search space of FOA to a three-dimensional search space, which can effectively restrain FOA algorithms from getting trapped in local optimal solutions and improve the global search power and convergence rate.

(3) The FOA algorithm based on 3D-LSCCM was used to search for the optimal combination value of the key parameter $[K, \alpha]$ of VMD to ensure that the VMD algorithm could adaptively obtain the best decomposition performance.

(4) The fault signal is decomposed using the optimal VMD method to obtain several IMF components, which are selected to include shocks based on the mean kurtosis criterion for the IMF components. The selected IMF components are then reconstructed to extract fault characteristic frequencies more efficiently. Finally, the experimental results show that the combined CTEO and optimal VMD approaches have excellent performance and advantages in extracting early fault characteristics of mechanical devices.

(5) In the actual operation of large, low-speed, and heavy-duty mechanical devices, in addition to the low-speed and large load of the mechanical devices themselves, there may be some intermittent operating characteristics. For this condition, the validity of the proposed method needs to be further verified.

**Author Contributions:** Conceptualization, B.X., F.Z. and H.L.; methodology, H.L.; software, B.X.; validation, P.H. and H.L.; formal analysis, P.H.; investigation, F.Z.; resources, F.Z.; project administration, F.Z.; data curation, H.L. and P.H.; writing—original draft preparation, H.L.; writing—review and editing, H.L. and P.H.; funding acquisition, B.X. All authors have read and agreed to the published version of the manuscript.

**Funding:** This research was supported by the National Natural Science Foundation of China (grant no. 51975433, 51975430), the Research Project of Hubei Provincial Department of Education (grant no. B2022203).

**Institutional Review Board Statement:** Not applicable.

**Informed Consent Statement:** Not applicable.

**Data Availability Statement:** Experimental data came from the authors' own collection and calculation.

**Acknowledgments:** The authors acknowledge the editors and reviewers for their constructive comments and all the support on this work.

**Conflicts of Interest:** The authors declare no conflict of interest.

## Appendix A

| **Algorithm A1:** Improved Chaotic FOA (ICFOA) |
| --- |

**Input:**

    $N_S$: number of individuals in fly swarm

    $D$: dimension of the search space

    $S$: smell factor

    $\theta$: control parameters of logistic-sine mapping

    $B$: borders of the search space

    $Ic_{1max}$: maximum number of generations that the map and compass operation is carried out.

    $Ic_{2max}$: maximum number of generations that the landmark operation is carried out.

**Initialize:**

    $Ic_{1max}$ = T1, $Ic_{2max}$ = T2, $N_S$ = p, $D$ = d, $\theta$ = 0.99, $S$ = random, $B$ = [b1, b2]

    **for** $Ns$ = 1 **to** $p$ **by** 1

    **do**

        **for** $d$ = 1 **to** $D$ **by** 1

        **do**

$$X_{axis}^{Ns} = B \times random(1,d)$$
$$Y_{axis}^{Ns} = B \times random(1,d)$$
$$Z_{axis}^{Ns} = B \times random(1,d)$$
$$Dist_{Ns} = \sqrt{\left(X_{axis}^{Ns}\right)^2 + \left(Y_{axis}^{Ns}\right)^2 + \left(Z_{axis}^{Ns}\right)^2}$$
$$S_{Ns} = 1/Dist_{Ns}$$

        **end for**

    **end for**

    $S_p = S_{Ns}$, $I_c = 1$

    $f(S_p)$ = fitness $(S_p)$

    $S_{gbest}$: = arg min $[f(S_p)]$

**Smell operations:**

    **for** $Ic$ = 1 **to** T1 **do**

        **for** $Ns$ = 1 **to** $p$ **do**

            **while** $S_{Np} > B$ **do**

$$\begin{cases} X_{Ns} = X_{axis}e^{-R \times I_c} + a\widetilde{X}_{Ns} - b \\ Y_{Ns} = Y_{axis}e^{-R \times I_c} + a\widetilde{Y}_{Ns} - b \\ Z_{Ns} = Z_{axis}e^{-R \times I_c} + a\widetilde{Z}_{Ns} - b \end{cases}$$

$$\begin{cases} \widetilde{X}_{Ns+1} = \sin((4\theta \times \widetilde{X}_{Ns}(1 - \widetilde{X}_{Ns}) + \sin(\pi \times \widetilde{Y}_{Ns})(1 - \theta))\pi) \\ \widetilde{Y}_{Ns+1} = \sin((4\theta \times \widetilde{Y}_{Ns}(1 - \widetilde{Y}_{Ns}) + \sin(\pi \times \widetilde{X}_{Ns+1})(1 - \theta))\pi) \\ \widetilde{Z}_{Ns+1} = \sin((4\theta \times \widetilde{Z}_{Ns}(1 - \widetilde{Z}_{Ns}) + \sin(\pi \times \widetilde{Y}_{Ns+1})(1 - \theta))\pi) \end{cases}$$

           **end while**

        **end for**

        evaluate $S_{Ns}$, update $X_{axis}$, $Y_{axis}$, $Z_{axis}$ and $S_{gbest}$

    **end for**

**Vision operations:**

    **for** $I_c$ = 1 **to** T2 **do**

        **while** $S_p > B$ **do**

$$N^{I_c} = \frac{N^{I_c-1}}{2}$$
$$S_{Ns} = S_{Ns}^{Ic-1} + random(S_{center}^{Ic-1} - S_{Ns}^{Ic-1})$$

        **end while**

        evaluate $S_{Ns}$, update $X_{axis}$, $Y_{axis}$, $Z_{axis}$ and $S_{gbest}$

    **end for**

**Output:** $S_{gbest}$.

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
