# Peer review of "Mechanical Incipient Fault Detection and Performance Analysis Using Adaptive Teager-VMD Method"

_applsci, doi:10.3390/app13106058_

Round 1

Reviewer 1 Report

There are certain recommendations to improve the scientific soundness and quality of manuscript. Comments are as follows:

1. Length of manuscript is very lengthy with unnecessary descriptions of methodology. Suggested to strengthen the results and discussion section.

2. Initially, authors developed a mathematical model and discussed chaotic plot. Later experimentation was done, and envelope spectrum was plotted and discussed. Connectivity between different sections and the aim of proposed methodology is highly uncertain. Kindly re-organize the whole manuscript.

3. When signal processing techniques like Wavelet and VMD is applied on signal, challenges related to mother wavelet selection and suitable IMF selection are left and unaddressed by authors. Kindly refer following journals and add description in revised version :

a. https://ieeexplore.ieee.org/abstract/document/8038631

b. https://www.mdpi.com/2076-3417/12/13/6470

4. Section 2.5 discussed fault feature extraction. Authors should include details what fault features are extracted and where it is utilized.

5. Conclusion section should be re-written. Findings should be highlighted precisely.

6. The envelope spectrum shown in Fig. 20, 21 and elsewhere can be represented in a better way. Please refer literature

Minor editing of English language required

Reviewer 2 Report

It is well known that most algorithms available to be applied to signal filtering tend to ignore useful information about the operation of rotating machines, especially if this information is verified at low amplitudes, which frequently occurs in machines that operate under high loads and low speeds. As such, the relevance and usefulness of the study presented here is unquestionable. There are, however, several weaknesses in the manuscript, which I consider important. In my opinion if observations below are implemented, the manuscript would be significantly improved:

1)      The manuscript requires a revision by a native English speaker: Please revise;

2)      Title should be rethink since roller bearings is only one possible application: Please revise;

3)      Keywords should be reconsidered in order to improve indexing capabilities and to accurately reflect the paper contributions: Please revise;

4)      The purposes are not explicitly presented and discussed (this is evident in the abstracts and conclusions: Please revise;

5)      The abstract and introduction do not provide clear results and recommendations drawn from the study conducted: Please revise;

6)      Sections and subsections (paper organization/structure) should be reconsidered in order to promote a easy reading and understanding of the paper (2 difficult tasks in the version now submitted): Please revise;

7)      Could be the contents of some sections shortened (see previous point)? Please revise;

8)      Cannot be algorithms provided as supplementary content (see previous points)? Please revise;

9)      Figures formatting: Please revise.

The manuscript requires a revision by a native English speaker

Round 2

Reviewer 1 Report

I do not having any further comments as authors addressed comments and modified manuscript accordingly.

Minor editing of English language required.